# Effects of Sex, Age, and Body Size on Flight Performance of *Monochamus alternatus* (Coleoptera: Cerambycidae), a Vector of Pine Wood Nematodes, Using Flight Mills

**DOI:** 10.3390/insects16050444

**Published:** 2025-04-23

**Authors:** Jong-Kook Jung, Chayoung Lee, Beomjun Jang, Youngwoo Nam

**Affiliations:** 1Department of Forest Environment Protection, Kangwon National University, Chuncheon 24341, Republic of Korea; jkjung@kangwon.ac.kr; 2Forest Insect Pests and Diseases Division, National Institute of Forest Science, Seoul 02455, Republic of Korea; ishurain@kangwon.ac.kr (C.L.); jbj2729@nie.re.kr (B.J.)

**Keywords:** pine wilt disease, vector insects, pine wood nematode, *Monochamus* beetles, flight capability

## Abstract

The pine wood nematode (PWN), *Bursaphelenchus xylophilus* (Steiner and Buhrer, 1934; Nickle, 1970), transmitted by *Monochamus* beetles causes pine wilt disease. It is a serious forest pest worldwide. In Korea, *Monochamus alternatus* (Hope, 1843) is an important vector of PWN. The flight abilities of *Monochamus* beetles are valuable for understanding the spread pattern of trees killed by the PWN. Therefore, this study investigated the effects of sex, adult age, and morphological traits of *M. alternatus* on its flight performance. Male adults showed better performance in terms of flight distance traveled compared to females. Analysis of the morphological traits showed that larger beetles with a larger thorax have a greater flight range. In addition, sex-related morphological differences play a crucial role in flight performance, particularly in determining flight distance and speed. Consequently, *M. alternatus* exhibits different flight patterns depending on sex, suggesting the necessity for sex-specific monitoring and control strategies to effectively manage vectors of pine wilt disease.

## 1. Introduction

*Monochamus* spp. (Coleoptera: Cerambycidae) are vectors of pine wood nematode (PWN), *Bursaphelenchus xylophilus* (Steiner and Buhrer, 1934; Nickle, 1970), a serious pest and the cause of pine wilt disease in East Asia and parts of Western Europe, [1,2,3,4]. The PWN was first discovered in Korea in 1988, affecting trees on Mt. Geumjeong in Busan [5], and since then, it has spread to most parts of South Korea [6,7]. Since 1988, approximately 16 million pine trees have been infected by the PWN and subsequently died [6]. In 2024, approximately 0.9 million pine trees died, with an estimated budget of $79.4 million having been allocated for PWN control efforts [6].

In Korea, *Monochamus alternatus* (Hope, 1843) and *M. saltuarius* (Gebler, 1830) are known vectors of the PWN. In particular, *M. alternatus* is primarily distributed in East Asia (including China, Japan, and Korea) and Southeast Asia (including Laos, Taiwan, and Vietnam) [8]. It is a major vector of PWN in Korea [1], Japan [9], and China [10]. In Korea, the distribution of two vectors is different. *Monchamus saltuarius* is mainly found in the central regions of South Korea, while *M. alternatus* is predominantly distributed in southern regions [11]. This beetle primarily attacks pine trees, particularly *Pinus densiflora* (Siebold and Zucc.) and *P. thunbergii* (Parl.) in Korea and Japan [1,9,12,13,14] and *P. massoniana* (Lamb.) in China [15,16]. Regarding the life cycle of *M. alternatus*, female adults lay eggs after making wounds on the bark of pine trees, and hatched larvae feed on the cambium layer and the outer part of the sapwood beneath the bark as they grow [17]. From August to October, the larvae bore into the sapwood to create pupal chambers where they overwinter as mature larvae [17]. After overwintering, the larvae pupate between late April and mid-June. Newly emerged adults escape from the pupal chambers from mid-May to early August [17]. Adults feed for about two weeks before reaching sexual maturity; this feeding is called maturational feeding [18].

Estimating the dispersal ability of insects has been a challenging task. Typically, mark-release-recapture experiments are conducted in the field using multi-funnel traps with pheromone lures [19,20]. In the lure, monochamol, known as an aggregation-sex pheromone, and kairomone, a mixture of several chemical compounds including α-pinene, β-pinene, and ethanol [21,22], are used to attract male and female adults of *Monochamus* spp. In the laboratory, flight mills are used to estimate the dispersal ability [23]. Field experiments often demonstrate a consistent decline in recapture rates with increasing distance from the release point. However, the low recapture rate of released individuals may hinder estimates of dispersal capability in the field [20,24]. In contrast, flight mills allow continuous flight experiments until the stored energy resources of each beetle are exhausted, making it possible to estimate potential flight capabilities such as flight distance, time, and speed. Thus, laboratory experiments using flight mills can indirectly estimate actual dispersal abilities under field conditions [23].

The flight abilities of *Monochamus* beetles are valuable for understanding the spread pattern of trees killed by the PWN. However, few studies have been conducted, especially for *M. alternatus*. Studies on the flight capabilities of *Monochamus* beetles include those on *M. carolinensis* (Olivier, 1792) in North America [25,26,27], *M. galloprovincialis* (Olivier, 1795) in Europe [23], *M. saltuarius* in Asia and Europe [28], and *M. alternatus* in Asia [29,30]. For *M. saltuarius*, the total flight distance during the beetle’s lifecycle was found to be an average of 1.93 km for females and 2.71 km for males [28]. This distance is comparable to a maximum annual spread distance of 1.48 km for trees killed by the PWN [31], indicating a similarity between total flight distance and field spread of pine wilt disease.

For *M. alternatus*, population density increases rapidly in environments with many dead pine trees, such as areas affected by forest fires [32], potentially accelerating the spread of pine wilt disease. However, only two studies have been conducted in Japan, one recording flight time across the entire lifespan [29] and the other reporting total flight distance [30], making it challenging to estimate flight distance and speed across different adult ages (i.e., post-emergence days) and the entire lifespan. Therefore, this study investigated the effects of sex, adult age, and morphological traits of *M. alternatus* on its flight performance such as flight distance, time, and speed.

## 2. Materials and Methods

### 2.1. Study Insects

In the spring of 2019, logs of Korean pine, *Pinus koraiensis* (Siebold and Zucc.) (100 cm in length, 15~20 cm in diameter) were used for oviposition by *M. alternatus*. Beetles were reared indoors and provided by Osangkinsect corporation. In May 2020, a total of 42 adult beetles (25 females and 17 males) that emerged from those logs in large, mesh-covered cage in Hongneung experimental forests in the National Institute of Forest Science were carefully collected and individually placed in insect rearing containers (8 cm in diameter and 12 cm in height). They were provided with pine shoots, insect-rearing jelly, and water-soaked cotton to prevent drowning. All adult beetles were kept in a 25 ± 1 °C constant temperature facility with a 16L:8D light-dark cycle during flight experiments at the Forest Insect Pests Laboratory of the National Institute of Forest Science. In the constant temperature facility, the relative humidity was maintained between 60 and 70%.

To measure the morphological traits of *M. alternatus* adults, photographs were taken for each insect from dorsal and lateral views immediately after death. Using ImageJ software version 1.46r (U.S. National Institutes of Health, Bethesda, MD, USA), body length (from clypeus of head to apex of abdomen), body width (elytral shoulder), body height (thickest part of the mesothorax), thorax width (pronotum), thorax height (pronotum), hindwing areas, and wing loads were measured (Figure A1). To calculate the wind loads, dry weight was divided by hindwing area. All dead bodies of *M. alternatus* were immediately dried in an oven (DS-80-1, Dasol Scientific, Hwaseong, Republic of Korea) for one day at 60 °C and their dry weight was measured using an electronic scale (ML204/01, Mdeeler-Toledo, Greifensee, Switzerland).

### 2.2. Measurement of Flight Distance, Time, and Speed

From May to July 2020, flight experiments were conducted over approximately eight weeks at the Forest Insect Pests Laboratory of the National Institute of Forest Science using custom-built flight mills [28] (Figure A2). A set of flight mills was placed and operated in independent rooms to minimize disturbances from other factors. All experiments were conducted during the daytime, based on the ecology of *Monochamus* beetles [22]. Each *M. alternatus* was individually tethered to a flight mill by bending an insect pin into an L-shape and securing it with a glue gun, after which flight distance was measured. Adult beetles were subjected to flight experiments once a week. Pine branches (5 cm long and 1 cm diameter) and insect jelly were provided for six days prior to the experiments to allow for energy accumulation. The pine branch was replaced every 2~3 days. As *M. saltuarius* [28] and *M. alternatus* [29] are known to fly very little immediately after emergence, adults were allowed to feed for three days post-emergence. The first experimental session was conducted on the fourth day after emergence. All flight experiments continued individually until the insects died.

A preliminary test using some other *M. alternatus* adults, which were not the same individuals as those used in our flight experiment, showed stop-and-long flight patterns repeatedly for two hours per session. By the end of each session, their speed dramatically decreased until they eventually stopped. Therefore, each flight experiment lasted for two hours. If an insect started flying late and continued beyond the two-hour session time, the experiment was extended until the insect ceased flying.

Of a total of 42 insects, most died after the sixth session experiment. A total of 13 beetles and 5 beetles were used to conduct the seventh and eighth sessions, respectively. During flight experiments, the distance and time of flight were automatically measured and saved in .csv file format using LabVIEW 2018 version 18.0f2 (National Instruments Corporation, Seoul, Republic of Korea), which was then used to calculate the distance (m), time (s), and speed (m s^−1^) per individual and session. If an insect flew only a very short distance (under several m per each flight) due to discomfort or unsuitable conditions, it was not considered a valid flight. Thus, flight behavior was analyzed when the total flight distance over two hours exceeded 100 m, excluding cases where free flight was nearly impossible due to very low speed. In this study, the average flight speed of all short distance flyers with a flight distance below 100 m was 1.2 m s^−1^. Many of them flew at speeds below 1.0 m s^−1^. Although some beetles showed high speeds, their flight distances were too short, i.e., below several m, and showed abnormal flight patterns such as discontinuous flight during each two-hour session in addition to short flight distances. Therefore, only sessions in which the total flight distance exceeded 100 m within the two-hour session were considered valid for calculating the distance, time, and speed for that session.

### 2.3. Data Analyses

To compare morphological traits between females and males, a two sample *t*-test assuming equal variances was conducted. To compare the flight ability (flight distance, time, and speed) of *M. alternatus* based on sex (female and male) and age (weeks passed from the first experimental session of the experiment after emergence to the eighth session), a linear mixed model (LMM) analysis was conducted. For LMM, the data for flight ability did not meet the assumption of normal distribution, and as such, log-transformed data for flight ability were used. Pearson’s correlation coefficient was used to determine the relationship among morphological traits.

Additionally, to analyze the relationship between measured morphological traits (dry weight, body length, thorax width and height, wing area, elytral width and height, and wing load) and flight performance (flight distance, speed, and time), principal component analysis (PCA) was performed. Only individuals with a flight distance exceeding 100 m per session were included in the analysis, which consisted of 22 females and 16 males. As a scree plot revealed an elbow phenomenon where the change in eigenvalues became negligible beyond the fifth principal component (PC5), we only used data up to the fourth principal component (PC4), as this explained more than 75% of the total variance. Correlation analysis was conducted to examine relationships among the principal components (PC1 to PC4) and each variable (flight distance, flight speed, dry weight, body length and width, thorax height and width, and wing area and load).

All data analyses were performed using R version 4.4.0 [33]. Student’s *t*-test was used to compare morphological traits between males and females. LMMs were conducted using the ‘lmer’ function from the ‘lme4’ and ‘lmerTest’ packages and the ‘anova’ function from the ‘car’ package. Among the morphological traits, correlation coefficients and *p*-values were calculated using the ‘rcorr(as.matrix)’ function from the ‘Hmisc’ package. Principal component analysis was performed using the ‘prcomp’ function. Biplots were created using the ‘ggbiplot’ function from the ‘ggbiplot’ package. Biplots were constructed by combining PC1 to PC4. For correlation analysis, the coordinates of each experimental insect were identified using the ‘get_pca_ind’ function from the ‘factoextra’ package. Correlation coefficients and *p*-values among principal components (PC1 to PC4) and each variable were calculated using the ‘rcorr(as.matrix)’ function from the ‘Hmisc’ package.

## 3. Results

### 3.1. Summary of Morphological Traits and Flight Experiments

The observed morphological traits of the *M. alternatus* tested on flight mills are presented in Table 1. Dry weight and body length did not significantly differ between males and females. Thorax width (*t* = 3.59, *p* < 0.001) and height (*t* = 3.95, *p* < 0.001) were significantly larger in males than in females. However, elytral length (*t* = −32.35, *p* < 0.001), width (*t* = −2.43, *p* < 0.05) and hindwing area (*t* = −2.69, *p* < 0.05) were significantly larger in females than in males. In addition, wing load (the load per unit wing area) was higher in males than in females (*t* = 2.16, *p* < 0.05).

As a result of flight experiments conducted on a total of 42 individuals, it was confirmed that 22 out of 25 females and 16 out of 17 males flew more than 100 m in each session (Table 2). Three females and one male consistently flew less than 100 m (average 13.1 m, ranging from a minimum of 0.6 m to a maximum of 78.0 m) during the experiment period. They died between the third and fifth sessions.

During the total of eight sessions, excluding dead beetles, females attempted a total of 140 flights, of which 84 flew more than 100 m, while males attempted 103 times, of which 67 flew more than 100 m. In the final sessions before their death, some beetles flew only a few hundred meters, while most individuals did not fly at all.

### 3.2. Change in Survival Rates Across Lifespan

The survival rate of females was over 80% up to the fourth to fifth sessions. It then gradually decreased to 60% in the sixth session (by which time 10 out of 25 beetles had died), 24% in the seventh session (19 beetles had died), and 8% in the eighth session (23 beetles had died) (Figure 1a). A similar trend was observed in males, with a survival rate of 94.1% in the fourth session. However, the survival rate gradually decreased to 76.5% in the fifth to sixth sessions (4 out of 17 beetles had died), 41.2% in the seventh session (10 beetles had died), and 17.6% in the eighth session (14 beetles had died). After the eighth session, no individuals were alive in either group.

### 3.3. Total Flight Distance, Time, and Speed

The total flight distance of individuals that flew more than 100 m per session showed that females flew an average of 6.65 ± 0.75 km (S.E.) for 59.6 ± 31.38 min, while males flew an average of 9.89 ± 1.98 km for 85.8 ± 69.37 min (Table 2). Throughout the entire experimental sessions, males flew significantly farther than females (LMMs; *d.f.* = 143.95, *F* = 4.12, *p* = 0.044) (Table 3). However, flight time (LMMs; *d.f.* = 145.56, *F* = 3.17, *p* = 0.077) and average speed (LMMs, *d.f.* = 141.21, *F* = 1.51, *p* = 0.222) showed no significant difference between females and males.

In each flight session, females and males flew an average of 1.76 ± 0.15 km (13.8 ± 4.50 min) and 2.36 ± 0.21 km (17.7 ± 6.93 min), respectively. There was no significant difference in flight distance (LMMs; *d.f.* = 136.67, *F* = 1.48, *p* = 0.181) or flight time (LMMs; *d.f.* = 138.02, *F* = 1.19, *p* = 0.311) across flight sessions, i.e., ages measured as weeks after adult emergence (Table 3). The individual that flew the farthest in a single session was a male (7.57 km), surpassing the farthest flying female (7.09 km) (Table 2). In terms of total flight distance, females flew up to 15.3 km, while males flew up to 29.0 km. The flight speed averaged 1.8 ± 0.16 m s^−1^ and 1.9 ± 0.22 m s^−1^ for females and males, respectively, with a range of 1.32 m s^−1^ to 2.34 m s^−1^ (Table 2). Flight speeds across ages (LMMs; *d.f.* = 134.98, *F* = 1.72, *p* = 0.109) were not significantly different (Table 3).

Regarding the average flight distance measured at one-week intervals, beginning one week after adult emergence, both males and females maintained a consistent flight distance from the first to the sixth sessions (Figure 1b). However, from the seventh session onward, as mortality rates increased, the average flight distance dramatically decreased.

Among the 38 individuals that flew, 18 females (76.0%) recorded flight distances within 10 km, while 9 males (56.3%) flew within the same range. One female (4.5%) and three males (18.8%) recorded flight distances exceeding 15 km. Except for one individual of each sex, all individuals flew more than 1 km. Regardless of sex, it appeared that the total flight distance for each beetle was not influenced by total time, which ranged from 4 h (2 sessions) to 16 h (8 sessions) for individuals tested on flight mills throughout their lifespan (males: *r* = 0.34, *p* = 0.121; females: *r* = 0.22, *p* = 0.420).

### 3.4. Principal Component Analysis

Regarding the eigenvalue for each component from the principal component analysis (PCA), PC1 had the highest eigenvalue of 1.794, explaining 29.3% of the total variance, followed by PC2 (eigenvalue 1.449, explaining 19.1%), PC3 (eigenvalue 1.306, explaining 15.5%), and PC4 (eigenvalue 1.137, explaining 11.8%) (Table 4). The cumulative variance explained by PC1 to PC4 was 75.6%, with the remaining 24.4% explained by PC5 to PC11.

According to their morphological traits, *M. alternatus* adults were arranged across PC1, particularly body size-related traits, including body length (*r* = −0.82, *p* < 0.001), elytral width (*r* = −0.78, *p* < 0.001), dry weight (*r* = −0.77, *p* < 0.001), wing load (*r* = −0.64, *p* < 0.001), and thorax width (*r* = −0.50, *p* = 0.002) in addition to flight time (*r* = 0.52, *p* = 0.001) and distance (*r* = 0.48, *p* = 0.002) (Figure 2, Table 4). *M. alternatus* adults were distinctly separated into two groups of females and males by PC2, which was also related to other morphological traits such as thorax width (*r* = 0.69, *p* < 0.001), thorax height (*r* = 0.49, *p* = 0.002), and wing area (*r* = −0.39, *p* = 0.017) in addition to flight distance (*r* = 0.73, *p* < 0.001) and flight time (*r* = 0.72, *p* < 0.001) (Figure 2, Table 4).

PC3 and PC4 were also important components for flight distance (*r* = −0.33, *p* = 0.046) and flight speed (*r* = −0.40, *p* = 0.013), respectively (Table 4), while wing area (*r* = −0.85, *p* < 0.001) and thorax height (*r* = 0.72, *p* < 0.001) were the most influenceable traits for PC3 and PC4, respectively.

## 4. Discussion

### 4.1. Flight Capability of M. alternatus Tethered on Flight Mills

This study examined the flight capabilities in terms of the flight distance, time, and speed of *M. alternatus* adults from the Korean population. Previously, a Japanese population study of *M. alternatus* has shown that five-day-old adults could fly for more than 16 min for females and 20 min for males [29]. In this study, relatively less flight time and distance (i.e., average 13.7 min for 1.45 km and 16.1 min for 1.90 km) were recorded in the first experimental session (i.e., four-day-old adults) for females and males, respectively. A later study considered this species to be a short-distance flyer, typically traveling less than 4 km over their lifespan [30]. Therefore, different experimental conditions and beetle status might hinder direct comparisons between the results of our study and those of previous studies for *M. alternatus* [29,30].

This study showed the intermediate flight capability of *M. alternatus* between *M. saltuarius* (averaging 1.9 to 2.7 km) [28] and *M. galloprovinciallis* (averaging 15.6 to 16.3 km) [23]; therefore, we assumed relationships between body size and flight capability. In this study, the tested *M. alternatus* adults had an average body length of 25.6 mm (ranging from 21.6 to 29.1 mm), which may be one of the key factors influencing flight capability. When comparing body sizes across species, *M. saltuarius* is the smallest, with a body length ranging from 12.8 mm to 21.4 mm, whereas *M. galloprovincialis* is the largest, ranging from 14 mm to 35 mm. In *M. galloprovincialis*, a significant positive correlation between body weight at emergence and total flight distance was reported [23], indicating that larger individuals exhibit greater dispersal potential. Similarly, *M. saltuarius* individuals with larger body sizes could fly significantly longer distances [28]. These findings further support the hypothesis that body size is a critical determinant of dispersal ability in *Monochamus* species, suggesting that flight capacity is strongly influenced by this factor.

In addition, the *M. alternatus* tested in this study were reared indoors on *P. koraiensis* logs, which may have resulted in differences in body size and flight performance compared to individuals that feed on and emerged from *P. densiflora* or other host tree species under natural conditions. In fact, *M. alternatus* is native to the southern part of South Korea. Therefore, it would be inappropriate to assume that the flight capability observed in this study directly reflects that of wild populations. In natural conditions, it was confirmed that the average body length of *M. alternatus* that may have emerged from dead pine trees, i.e., *P. densiflora* or *P. thunbergii*, collected from forests in Jeju Island, located in the southern part of the Korean Peninsula, was smaller, with females averaging 20.97 mm and males 19.93 mm [34]. This size difference is notably greater than that observed in our study. Such differences suggest that various factors, including rearing conditions, host tree species, and nutritional status differences, may influence not only body size but also flight performance. As such, caution should be exercised when extrapolating the findings of this study to natural populations, and potential environmental influences should be considered when interpreting the results.

Nonetheless, the flight capability of *M. alternatus* based on flight mill experiments plays a crucial role in understanding the dispersal mechanisms of pine wilt disease. In this study, we measured the flight distance of *M. alternatus* using flight mills, revealing that females had an average flight distance of 1.76 km per session while males averaged 2.36 km. The flight ability of *M. alternatus* is particularly relevant when compared with the estimated spread rate of pine wilt disease-infected trees in Korea. Previous studies have reported that the spread rate of pine wilt disease-infected trees is 1.48 km per year [31], while a more recent study estimated an average annual spread rate of 13.8 km, ranging from 9.4 to 24.6 km, between 2017 and 2021 [7]. Given that the maximum total flight distance recorded for female *M. alternatus* in this study was 15.3 km, these results suggest that this species is capable of long-distance dispersal under natural conditions.

### 4.2. Sex-Dependent Flight Performance of M. alternatus

This study found that males flew farther than females across most experimental sessions. This finding contrasts with previous studies on *M. saltuarius* [28], *M. alternatus* [29], and *M. galloprovincialis* [23], which reported no significant sex-based differences in flight performance. The discrepancy with previous studies might be attributed to morphological differences between female and male adults. Although we did not measure weight immediately after emergence, the males in our study had a greater thorax width and height, while females exhibited larger elytral length, width, and hind wing areas, indicating distinct morphological characteristics between sexes. Males also exhibited higher wing loading than females, and thus, the lower wing loading of the females may have contributed to their enhanced maneuverability [35]. However, as dry weight after the insects died was used to calculate wing loads in this study, this parameter should be reassessed. *Monochamus* beetles are known to be iteroparous, i.e., they reproduce multiple times throughout their lifespan [17], and females are likely to retain eggs throughout their oviposition periods, which may increase their wing loads. Thus, the wing loading of female may have been influenced by internal physiological factors.

Flight speed also varied by sex in *Monochamus* species. *M. galloprovincialis* had an average flight speed of 1.43 m s^−1^ for females and 1.35 m s^−1^ for males [23], while *M. saltuarius* flew at speeds of 1.08 to 1.30 m s^−1^ [28]. In this study, *M. alternatus* had the fastest flight speed among comparable *Monochamus* species, with males and females averaging 1.92 m s^−1^ and 1.86 m s^−1^, respectively. Without specific body size information from previous studies, it is difficult to clearly explain such interspecies differences in flight speed. Although the shape of flight muscle, as well as the width and height of the elytral shoulder and thorax (i.e., pronotum in this study) are crucial factors for estimating flight speed as a proxy for flight muscle mass, we did not measure them. Flight speeds could increase with increasing amount of flight muscle. Positive relationships between flight speed and thorax width and height were more pronounced in males than in females, suggesting that males allocate more resources to developing flight muscles. This investment may contribute to their higher flight speeds and longer flight distances. Overall, these findings suggest that sex-related morphological differences play a crucial role in flight performance, particularly in determining flight distance and speed. For example, in females, it appears that dispersal ability is important, as they are required to lay their eggs.

## 5. Conclusions

This study demonstrated that the flight ability of adult *M. alternatus* adults varies significantly depending on body size and sex. This information is fundamental and crucial for understanding the spread patterns of pine wilt disease. Although there are some differences between natural movement patterns and the flight mill experiment results, the average flight distance per single flight of *M. alternatus*, as measured in this study, was similar to the estimated annual spread distance of PWN-infected trees [31]. However, current estimates of the spread rate of PWN-infected trees in South Korea are based on nationwide data, which include not only *M. alternatus* but also other vectors such as *M. saltuarius*. In fact, in central Korea, where only *M. saltuarius* is distributed, the annual spread rate of PWN-infected tree is around 577 m to 666 m [36], which differs from the national average [31]. Therefore, developing a pine wilt disease spread prediction model that considers the flight abilities of different vector species is expected to improve the accuracy of spread predictions.

Additionally, because adult *M. alternatus* exhibit different flight patterns based on sex, it is necessary to apply sex-specific monitoring and control techniques for managing pine wilt disease vectors. For females, they are expected to move to nearby trees for maturational feeding and oviposition, and they may contribute to the short-distance spread of PWN during maturational feeding and oviposition. Therefore, methods such as placing fallen trees within forests to induce oviposition or using pheromone traps for mass-capture and to prevent long-distance movement could be implemented to ensure that newly emerged adults from infected trees do not disperse far [21,37,38]. This study found that both male and female adults of *M. alternatus* could fly up to approximately 2 km in a single flight. Therefore, it is crucial to designate a 2 km radius around initially detected infected trees as a focal zone for monitoring and control. This approach, considering the flight characteristics of *M. alternatus*, is expected to contribute to a more effective suppression of pine wilt disease spread.

## Figures and Tables

**Figure 1 insects-16-00444-f001:**
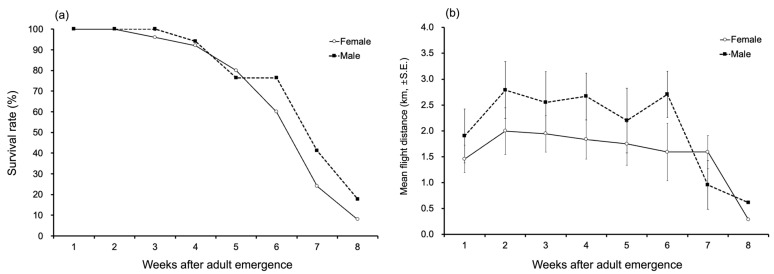
Changes in (**a**) survival rates and (**b**) flight distances of female and male *Monochamus alternatus* adults according to age (weeks after emergence). Only cases where individuals flew more than 100 m during the entire experiment were included in the analysis.

**Figure 2 insects-16-00444-f002:**
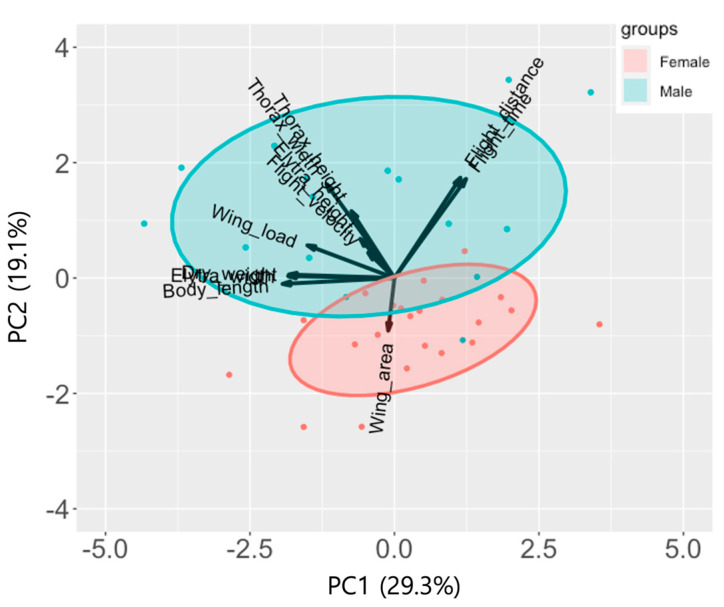
PCA biplot illustrating relationships among flight performance (flight distance and flight speed) and morphological traits of *Monochamus alternatus*. Different principal components (PCs) were used. PC1 was used as the *x* axis and PC2 was used as the *y* axis. The length of the arrow indicates the degree of correlation with the *x* or *y* axis. The direction of the arrow indicates positive or negative correlation.

**Table 1 insects-16-00444-t001:** Comparison of the morphological traits of the *Monochamus alternatus* tested on flight mills.

Morphological Traits	Male (*n* = 17)	Female (*n* = 25)	*t*	*d.f.*	*p*
Dry weight (mg)	153.59 ± 38.65	151.48 ± 41.34	0.17	40	0.8686
Body length (mm)	24.85 ± 3.76	25.29 ± 2.38	−0.46	40	0.6480
Thorax width (mm)	6.49 ± 0.52	5.17 ± 0.61	3.59	40	0.0009
Thorax height (mm)	5.99 ± 0.58	5.18 ± 0.7	3.95	40	0.0003
Elytral length (mm)	21.67 ± 2.14	23.31 ± 2.37	−32.35	40	0.0000
Elytral width (mm)	7.37 ± 0.76	8.01 ± 0.88	−2.43	40	0.0196
Hindwing area (mm^2^)	147.64 ± 30.36	173.74 ± 31.29	−2.69	40	0.0105
Wing load (mg/mm^2^)	1.12 ± 0.56	0.86 ± 0.16	2.16	40	0.0368

**Table 2 insects-16-00444-t002:** Summary of the test for flight distance, time, and speed of *Monochamus alternatus* tethered on flight mills. Total flight of fliers indicates the sum of flight behaviors, such as distance and time, in each session per individual, while flight speed is the average value of all experiments, i.e., flight data acquired from the first session to the eighth session per individual.

Sex	*n*			Flight Distance (km)		Flight Time (min.)		Flight Speed for Fliers (m s^−1^)	
	^a^ Total	^b^ Non- Fliers	^c^ Fliers	Mean ± S.E.	Range (Min.~Max.)	Mean ± S.E.	Range (Min.~Max.)	Mean ± S.E.	Range (Min.~Max.)
Total flight									
Female	25	3	22	6.65 ± 0.75	0.13~15.35	59.56 ± 31.38	1.62~131.50	1.86 ± 0.23	1.32~2.34
Male	17	1	16	9.89 ± 1.98	0.17~29.01	85.83 ± 69.37	1.39~241.07	1.93 ± 0.22	1.34~2.32
Flight per session									
Female	140	60	84	1.76 ± 0.15	0.10~7.09	13.84 ± 4.50	3.12~17.13	1.81 ± 0.16	1.52~2.05
Male	103	36	67	2.36 ± 0.21	0.12~7.57	17.75 ± 6.93	7.51~25.09	1.86 ± 0.22	1.36~2.04

^a^ The number of individuals in each flight test. ^b^ The number of individuals that did not fly while tethered on flight mills or flew under 100 m in each session. ^c^ The number of individuals that flew above 100 m in each session.

**Table 3 insects-16-00444-t003:** Results of linear mixed models (LMMs) on flight distance, time, and speed of *Monochamus alternatus* tethered on flight mills. Sex and session were treated as fixed effects, while beetle ID was treated as a random effect.

Dependent Variables	Fixed Effect	*Num d.f.*	*Den d.f.*	*F*	*p*
Flight distance	Sex	1	143.95	4.12	0.044
	Session	7	136.67	1.48	0.181
Flight time	Sex	1	145.56	3.17	0.077
	Session	7	138.02	1.19	0.311
Flight speed	Sex	1	141.21	1.51	0.222
	Session	7	134.98	1.72	0.109

**Table 4 insects-16-00444-t004:** Summary of principal component analysis. Correlation coefficient with *p*-value among parameters for flight and morphological traits and four principal components. *p*-values are indicated by asterisks: *, *p* < 0.05; **, *p* < 0.01; ***, *p* < 0.001.

	PC1	PC2	PC3	PC4
Eigenvalue	1.794	1.449	1.306	1.137
Proportion of variance	0.293	0.191	0.155	0.118
Cumulative proportion	0.293	0.484	0.639	0.756
Flight or morphological traits				
Flight speed	−0.19	0.19	0.28	* −0.40
Flight distance	** 0.48	*** 0.73	* −0.33	−0.30
Flight time	** 0.52	*** 0.72	−0.32	−0.26
Dry weight	*** −0.77	0.03	* −0.35	−0.31
Body length	*** −0.82	−0.04	−0.22	−0.28
Thorax width	** −0.50	*** 0.69	0.10	* 0.34
Thorax height	−0.31	** 0.49	0.06	*** 0.72
Wing area	−0.05	* −0.39	*** −0.85	0.10
Elytral width	*** −0.78	0.01	−0.23	0.11
Elytral height	−0.25	0.29	** −0.44	0.04
Wing load	*** −0.64	0.24	** 0.52	* −0.36

## Data Availability

The data presented in this study are available on request from the corresponding author due to ethical reasons.

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
