# Peer review of "Effects of Sex, Age, and Body Size on Flight Performance of Monochamus alternatus (Coleoptera: Cerambycidae), a Vector of Pine Wood Nematodes, Using Flight Mills"

_insects, 2025, doi:10.3390/insects16050444_

Round 1
Reviewer 1 Report
Comments and Suggestions for Authors
The manuscript entitled "Effects of sex, age, and body size on flight performance of Monochamus alternatus, a vector of pine wood nematodes, tethered by flight mills" represents an original scientific paper that studies the effects of sex, age, and body size on flight performance of Monochamus alternatus using flight mills. In general, the manuscript is well-written, with some small technical issues, especially in the Simple Summary, Abstract and Introduction. The specific comments are given in the attached document.

Author Response
Effects of sex, age, and body size on flight performance of Monochamus alternatus, a vector of pine wood nematodes, tethered by flight mills
Line 2: Proposition for the title: Effects of sex, age, and body size on flight performance of Monochamus alternatus (Coleoptera: Cerambycidae), using flight mills
- Response: Thank you for the comment. We corrected it.
Simple Summary: This chapter must be written as suggested, because it is not understandable.
Line 17: Bursaphelenchus xylophilus (Steiner & Buhrer, 1934) Nickle, 1970
- Response: Thank you for the comment. We corrected it.
Line 19: Monochamus alternatus Hope, 1843 is an important vector of PWN.
- Response: Thank you for the comment. We corrected it.
Lines 19-21: The sentence “Because flight abilities of Monochamus beetles are valuable for understanding the spread pattern of trees killed by the PWN, this study investigated effects of sex, adult age, and morphological traits of M. alternatus on its flight performance.” should be divided in two, to be more understandable, because this is simple summary. For example: The flight abilities of Monochamus beetles are valuable for understanding the distribution pattern of trees killed by PWN. Therefore, this study investigated the effects of sex, adult age, and morphological traits of M. alternatus on its flight performance.
- Response: Thank you for the comment. We corrected it.
Lines 22-24: These sentences should be changed: Male adults showed better performance in terms of flight distance traveled compared to females. Analysis of the morphological traits showed that larger beetles with a larger thorax have a greater flight range.
- Response: Thank you for the comment. We corrected it.
Abstract: The Abstract needs to be changed, because it is not understandable. This is a proposal for the change:
Estimating the dispersal ability of insects was a difficult task to understand the spread of the pine wood nematode Bursaphelenchus xylophilus (Steiner & Buhrer, 1934) Nickle, 1970 and infected trees. Therefore, in this study, the effects of sex, adult age, and morphological traits on flight performance (such as flight distance, time, and speed ) of Monochamus alternatus Hope, 1843 were investigated using flight mills. Over a period of 8 weeks, a total of 42 newly emerged adult beetles were tethered to flight mills once a week. Flight distance and time were recorded for 2 hours during each session. The cumulative flight distance of individuals that flew more than 100 m during each session was calculated. The results showed that females flew an average of 6.65 km, for 59.6 min, while males flew an average of 9.89 km, for 85.8 min. In a single flight experimental session, individuals of both sexes could fly up to approximately 2 km, and were found to fly at an average speed of 1.9 m s-1. In principal component analysis (PCA), the first and second principal components were highly correlated with the sex, morphological traits, and flight performance of M. alternatus. This study demonstrates that the flight ability of M. alternatus varies significantly according to body size and sex, which is fundamental and crucial for understanding the dispersal patterns of pine wilt disease.
- Response: Thank you for the comment. We corrected it.
- Introduction: This chapter is generally well written. However, some comments are added below.
Lines 48-51: The first sentence is too complicated and incomprehensible. So, I propose this change: Monochamus spp. (Coleoptera: Cerambycidae) are vectors of pine wood nematode (PWN), Bursaphelenchus xylophilus (Steiner & Buhrer, 1934) Nickle, 1970, a serious pest of pine forests in East Asia and parts of Western Europe, and causes pine wilt disease [1, 2, 3, 4].
- Response: Thank you for the comment. We corrected it.
Line 51: The PWN Line 52: ... and since then, it has spread to ...
- Response: Thank you for the comment. We corrected it.
Lines 53-54: The sentence “To control the spread of the PWN, hundreds of billions of Korean won are allocated annually.“ should be supported by a cited reference or deleted.
- Response: Thank you for the comment. Based on other reviewer comment, we deleted this sentence.
Line 57: Monochamus alternatus Hope, 1843 and M. saltuarius (Gebler, 1830)
- Response: Thank you for the comment. We corrected it.
Line 58: Delete the sentence “They belong to the Cerambycidae family of the order Coleoptera.“
- Response: Thank you for the comment. We corrected it.
Lines 60-61: It is a major vector of PWN in Korea [1], Japan [9] and China [10].
- Response: Thank you for the comment. We corrected it.
Lines 63-65: This beetle primarily attacks pine trees, particularly Pinus densiflora Siebold & Zucc. and P. thunbergii Parl. in Korea and Japan [1, 9, 6412, 13, 14] and P. massoniana Lamb. in China [15, 16].
- Response: Thank you for the comment. We corrected it.
Lines 71-72: Adults feed for about two weeks to reach sexual maturity, and this feeding is called maturational feeding [18].
- Response: Thank you for the comment. We corrected it.
Lines 77-78: Monochamus spp. ....flight mills are used to estimate...
- Response: Thank you for the comment. We corrected it.
Lines 88-89: ... M. carolinensis (Olivier, 1792) in North America [25, 26, 27], M. galloprovincialis (Olivier, 1795) in Europe...
- Response: Thank you for the comment. We corrected it.
Line 98: Please, unify the citation. Here, (Ito, 1982) has been used as a citation type, while everywhere else you used [ ]. (Ito, 1982) is [29]
- Response: Thank you for the comment. We corrected it.
- Materials and Methods: This chapter is well written. Some comments are added below.
Line 108: Pinus koraiensis Siebold & Zucc.
- Response: Thank you for the comment. We corrected it.
Line 131: Please provide some better pictures on Figure A2.
- Response: Thank you for the comment. Unfortunately, we don’t have high quality images, please understand.
Line 156: flight (Kwon et al., 2018). The sentence ’’This phenomenon was also observed in flight experiments with M. saltuarius (Kwon et al., 2018).“ should be deleted and/or moved to the discussion.
- Response: Thank you for the comment. We corrected it.
Lines 157-165: Some sentences should be moved to the Discussion. This part of the paragraph should be rewritten, without comments and explaining the results of other articles, etc. Only the procedure you used. Of course, you can cite Kwon et al. (2018) and their procedure.
- Response: Thank you for the comment. We corrected it.
- Results: This chapter is well written. Some comments are added below.
Line 197: It is better to start this chapter with the sentence: The results of the morphological traits of M. alternatus tested on flight mills are presented in Table 1. Dry weight and body length...
- Response: Thank you for the comment. We corrected it.
Line 199: The part of the sentence „which are related to flight muscles“ is part of the discussion and should be moved to the Discussion.
- Response: Thank you for the comment. We corrected it.
Line 204: (n=42) should be deleted from the Table 1 title. In Table 1, after Male, you should add (n=17), and after Female (n=15)
- Response: Thank you for the comment. We corrected it.
Line 215: You could have done Kaplan-Meier Survival analysis for the results on survival rate as a function of time.
- Response: Thank you for the comment. Your suggestion is better to understand our data about changes in survival rate across time, but current form might also be inferred survival rate patterns in alternatus. Please understand.
Lines 245-247: Please, check these results once again. Males flew more then 30% more time.
- Response: Thank you for the comment. In fact, we used log-transformed data, and I assumed that the difference may be not appeared in statistics.
Line 263: Table 3 is out of place.
- Response: Thank you for the comment. We corrected it.
- Discussion: This chapter is well written.
Line 329: which is mainly distributed in...
- Response: Thank you for the comment. We corrected it.
Line 356: ... replace „were“ at the end of the row with have
- Response: Thank you for the comment. We corrected it.
- Conclusions: This chapter is well written.
Line 380: delete the word data
- Response: Thank you for the comment. We corrected it.
Line 393: fallen trees
- Response: Thank you for the comment. We corrected it.
Line 417-419: This figure is fine. However, if it is possible to update this figure to be clearer.
- Response: Thank you for the suggestion. We changed the figure.
Reviewer 2 Report
Comments and Suggestions for Authors
The authors used flight mills to determine the dispersal capability of Monochamus alternatus beetles, which are an economically-important vector of the invasive pine wood nematode in Korea. They found that male beetles dispersed further than females. They also recorded higher distances than previous flight-mill studies of beetles of the same genus.
The introduction is informative and engaging but a little repetitive in places.
The methods and results are generally clear, but there are a couple of important queries.
The discussion was interesting and well written, but the structure could benefit from a little refining to create a clearer narrative.
For specific changes, see line-by-line changes below:
23-24 – “larger beetles with large-sized thorax” is a little awkward. Are those things connected or separate? You could say “…larger beetles and beetles with large thoraxes”, or if it is just one then “beetles with large-sized thoraxes”.
52 - Include the economic damage of PWN if the data are available.
56 – Do you mean Monochamus control effects? There are two values listed for the PWN budget: hundreds of billions KRW on line 53 and 79.4 million USD on line 56 - please clarify.
58 – 63 – repetition. You can combine these sentences as they say the same thing.
88 – 1. “on <the> flight capabilities”. 2. M.cacrolinensis needs italics
96 – “such as areas affected by forest fires [32] and other reasons,”
101-103 – reword and clarify: are the differences confirmed? Could you include a source for this?
147 – This is unclear. It reads as if you make exceptions for insects that do not fly immediately and extend the trial period until it stops flying. If so, then you need to reanalyse the data according to a consistent rule that is equal to all the insects. If lots of insects took a while to start flying, you could create a rule to analyse the two-hour period from when each insect started flying, but this would have to be the same for every insect and not just those which started flying late.
179 – fifth principle component = PC7? Is this right?
180 – Confusing, reword. Are you saying that you examined the first 4 PCAs because they explained >75% var?
198 – different
Table 1 – Please use at least 3 significant figures for estimates <10. For example: 5.2 ± 0.7 becomes 5.24 ± 0.73.
211-212 – Unclear language: were there a total of 140 flight experiments with females, of which 84 flew >100m to be analysed?
224-227 – move this information to the paragraph about flight distance.
Table 2 – 1. “cumulated” -> “cumulative”: do you mean the total flight distance across all sessions? If so, “Total flight” and “Flight per session” might be clearer. 2. Keep the number of decimal places/ significant figures consistent (consistent with table 1 too). There is an 85.8 for one metric, and 9.89 for another. 3. Remove “for fliers” from metric headings, e.g. “Flight distance for fliers”. You can mention in the table legend that only individuals flying >100m contribute towards the metrics.
Figure 1 – good, but why not also include graphs of speed and flight time?
Linear mixed models - 1. using linear models for flight distance, time and speed is concerning as I strongly suspect that the data are not normally distributed for all three parameters. Distance in particular will have a strong positive skew that will probably need a log transformation to correct. You did not mention in your methods if you tested for normality. The N numbers are not too excessive for a Shapiro-wilks test, but I would recommend plotting a histogram of your dependent variables and testing data transformations (most likely log, but try ^2) to see if they bring the data to a normal distribution. If not, a glmm with a gamma family can handle skewed data. Data interpretation will be more difficult as you will have to reverse transform your data for table 2 and figures, but it is necessary for the analysis. 2. You have already gone to the effort of measuring loads of parameters, why not include them as predictors in the regression? There is collinearity between sex and the physical attributes so you will have to include an interaction with sex for each physical attribute to create a separate estimate for each sex(I.e. distance ~ sex + dry weight + sex:dry weight). You will then have to do a model selection process – I would recommend removing variables that do not improve the model by AICc. This more in-depth analysis is not a necessity but could strengthen your conclusion about the effects of size on dispersal ability and enable you to make predictions for graphing.
255 – Choose either m/s or ms-1 and keep the style consistent throughout
256 – “velocities” -> “speeds”
Table 3: why are the d.f so high? If n is 1 per beetle per session, wouldn’t that be ~200 or so?
282 – There is no figure 3, do you mean figure 2?
Principle component analysis - I am not sure that this section adds much to your interpretation of the data. The PCA is confusing to interpret as different components correlate oppositely between different parameters. For example: distance and time correlate positively with thorax width and height in PC1, but negatively in PC2. To keep this analysis, I would improve the narrative, which unnecessarily repeats the data from table 5 without adding much. I would try to repeat less and add more that you can’t already see in table 5, particularly how the different components relate to sex. Unless there are distinct clusters in your data (separate from males vs females), I would consider removing the PCA because your previous analysis already covers the male vs female comparison quite well.
Tables 4-5 - You can combine these into one table. Move PCA’s 5-7 to the supplementary material if necessary.
304 – don’t include results in the discussion unless you are contrasting your experiment with results of other studies. This applies to later lines too.
305 – Do you mean an earlier study?
332 – Do you know if these dead pine trees in Jeju were P. koraiensis or P. densiflora?
350 – conditions
359 – wouldn’t higher wing loading reduce their maneuverability?
363 - what is the observed increase in female wing loading? This might contradict your previous point that males have higher wing loading.
365-376 – Nice section. Could tethering differences explain some of the difference in speed? The size of the glue attachment for instance could inhibit flight.
376-377 – could different dispersal rates matter more for males vs females? Is the dispersal ability of females more important as they are required to lay the eggs?
Figure A2 – Images could be improved
Comments on the Quality of English LanguageOverall, I don’t think this study needs much, but there were two methods that concerned me that will need clarifying and probably correcting before publication. These are the experiment time sampling from line 147 and part 1 of “Linear mixed models” (see above).
Author Response
The authors used flight mills to determine the dispersal capability of Monochamus alternatus beetles, which are an economically-important vector of the invasive pine wood nematode in Korea. They found that male beetles dispersed further than females. They also recorded higher distances than previous flight-mill studies of beetles of the same genus.
The introduction is informative and engaging but a little repetitive in places.
The methods and results are generally clear, but there are a couple of important queries.
The discussion was interesting and well written, but the structure could benefit from a little refining to create a clearer narrative.
For specific changes, see line-by-line changes below:
23-24 – “larger beetles with large-sized thorax” is a little awkward. Are those things connected or separate? You could say “…larger beetles and beetles with large thoraxes”, or if it is just one then “beetles with large-sized thoraxes”.
- Response: Thank you for the comment. In fact, this issue is also raised by Reviewer 1, so we re-write the sentence.
52 - Include the economic damage of PWN if the data are available.
- Response: Thank you for the comment. Because there were some repetitions, we delete this sentence.
56 – Do you mean Monochamus control effects? There are two values listed for the PWN budget: hundreds of billions KRW on line 53 and 79.4 million USD on line 56 - please clarify.
- Response: Thank you for the comment. To clarify, we deleted “To control the spread of the PWN, hundreds of billions of Korean won are allocated annually.”.
58 – 63 – repetition. You can combine these sentences as they say the same thing.
- Response: Thank you for the comment. In fact, we described specific status of two vector species in Korea. So, we added “In Korea, the distribution of two vectors is different.” before “Monchamus saltuarius is mainly found in central regions of South Korea, while alternatus is predominantly distributed in southern regions”.
88 – 1. “on <the> flight capabilities”. 2. M.cacrolinensis needs italics
- Response: Thank you for the comment. We corrected it.
96 – “such as areas affected by forest fires [32] and other reasons,”
- Response: Thank you for the comment. We corrected it.
101-103 – reword and clarify: are the differences confirmed? Could you include a source for this?
- Response: Thank you for the comment. We deleted it to clarify.
147 – This is unclear. It reads as if you make exceptions for insects that do not fly immediately and extend the trial period until it stops flying. If so, then you need to reanalyse the data according to a consistent rule that is equal to all the insects. If lots of insects took a while to start flying, you could create a rule to analyse the two-hour period from when each insect started flying, but this would have to be the same for every insect and not just those which started flying late.
- Response: Thank you for the comment and we agree we needed to meet standard in flight experiment. However, the proportion of these cases was not high (few cases), so we thought fliers in late also can be included in analysis. Please understand.
179 – fifth principle component = PC7? Is this right?
- Response: Thank you for the comment. We corrected it.
180 – Confusing, reword. Are you saying that you examined the first 4 PCAs because they explained >75% var?
- Response: Thank you for the comment. We re-write the sentence to clarify.
198 – different
- Response: Thank you for the comment. We corrected it.
Table 1 – Please use at least 3 significant figures for estimates <10. For example: 5.2 ± 0.7 becomes 5.24 ± 0.73.
- Response: Thank you for the comment. We corrected it.
211-212 – Unclear language: were there a total of 140 flight experiments with females, of which 84 flew >100m to be analysed?
- Response: Thank you for the comment. We re-write the sentence to clarify.
224-227 – move this information to the paragraph about flight distance.
- Response: Thank you for the comment. The paragraph is moved based on the suggestion.
Table 2 – 1. “cumulated” -> “cumulative”: do you mean the total flight distance across all sessions? If so, “Total flight” and “Flight per session” might be clearer.
- Response: Thank you for the comment. We corrected it.
Table 2 – 2. Keep the number of decimal places/ significant figures consistent (consistent with table 1 too). There is an 85.8 for one metric, and 9.89 for another.
- Response: Thank you for the comment. We corrected it.
Table 2 – 3. Remove “for fliers” from metric headings, e.g. “Flight distance for fliers”. You can mention in the table legend that only individuals flying >100m contribute towards the metrics.
- Response: Thank you for the comment. We corrected it.
Figure 1 – good, but why not also include graphs of speed and flight time?
- Response: Thank you for the comment. Flight time was highly related to distance traveled, while speed for fliers didn’t change across experimental sessions. So, we just presented flight distance across session.
Linear mixed models - 1. using linear models for flight distance, time and speed is concerning as I strongly suspect that the data are not normally distributed for all three parameters. Distance in particular will have a strong positive skew that will probably need a log transformation to correct. You did not mention in your methods if you tested for normality. The N numbers are not too excessive for a Shapiro-wilks test, but I would recommend plotting a histogram of your dependent variables and testing data transformations (most likely log, but try ^2) to see if they bring the data to a normal distribution. If not, a glmm with a gamma family can handle skewed data. Data interpretation will be more difficult as you will have to reverse transform your data for table 2 and figures, but it is necessary for the analysis.
- Response: Thank you for the valuable comment. In fact, we used log transformed data, while we didn’t write about it. So, we added sentences for the normality test and data transformation.
Linear mixed models - 2. You have already gone to the effort of measuring loads of parameters, why not include them as predictors in the regression? There is collinearity between sex and the physical attributes so you will have to include an interaction with sex for each physical attribute to create a separate estimate for each sex(I.e. distance ~ sex + dry weight + sex:dry weight). You will then have to do a model selection process – I would recommend removing variables that do not improve the model by AICc. This more in-depth analysis is not a necessity but could strengthen your conclusion about the effects of size on dispersal ability and enable you to make predictions for graphing.
Response: Thank you very much for your valuable comments and suggestions. Your recommendations are insightful and would certainly help deepen the understanding of our data. However, in our LMMs, we specifically focused on evaluating the effects of sex and session on flight distance, time, and speed. While your suggestion is highly appreciated and meaningful, it falls somewhat beyond the scope of the present work.
255 – Choose either m/s or ms-1 and keep the style consistent throughout
- Response: Thank you for the comment. We corrected it.
256 – “velocities” -> “speeds”
- Response: Thank you for the comment. We corrected it.
Table 3: why are the d.f so high? If n is 1 per beetle per session, wouldn’t that be ~200 or so?
- Response: Thank you for the comment. In fact, every d.f., especially for denominator d.f., were bellow 200. We corrected the Table (separate num d.f. and den d.f.).
282 – There is no figure 3, do you mean figure 2?
- Response: Thank you for the comment. We corrected it.
Principle component analysis - I am not sure that this section adds much to your interpretation of the data. The PCA is confusing to interpret as different components correlate oppositely between different parameters. For example: distance and time correlate positively with thorax width and height in PC1, but negatively in PC2. To keep this analysis, I would improve the narrative, which unnecessarily repeats the data from table 5 without adding much. I would try to repeat less and add more that you can’t already see in table 5, particularly how the different components relate to sex. Unless there are distinct clusters in your data (separate from males vs females), I would consider removing the PCA because your previous analysis already covers the male vs female comparison quite well.
- Response: Thank you for valuable comments. We were wonder to know the relationship between flight parameters and morphological traits in addition to separation between sexes. And we already tried to find out how different component relates to sexes in L281-285.
Tables 4-5 - You can combine these into one table. Move PCA’s 5-7 to the supplementary material if necessary.
- Response: Thank you for the comment. We corrected it.
304 – don’t include results in the discussion unless you are contrasting your experiment with results of other studies. This applies to later lines too.
- Response: Thank you for the comment. We deleted it and revised later ones with minimizing results.
305 – Do you mean an earlier study?
- Response: Thank you for the comment. Yes, we did.
332 – Do you know if these dead pine trees in Jeju were P. koraiensis or P. densiflora?
- Response: Thank you for the comment. Yes, we added species name ( densiflora or P. thunbergii) after it.
350 – conditions
- Response: Thank you for the comment. We corrected it.
359 – wouldn’t higher wing loading reduce their maneuverability?
- Response: Thank you for the comment. In fact, we discussed about females with lower wing loading. We revised the sentence.
363 - what is the observed increase in female wing loading? This might contradict your previous point that males have higher wing loading.
- Response: Thank you for the comment. We revised the sentence.
365-376 – Nice section. Could tethering differences explain some of the difference in speed? The size of the glue attachment for instance could inhibit flight.
- Response: Thank you for the comment. I’m not sure that relationship between the size of the glue attachment and flight behavior. In fact, saltuarius by Kwon et al. including me (J.K. Jung) and M. alternatus (this study) were tethered by same method (using small amount of glues by glue gun). There is a possible point that the glue gun uses heat to melt down glue, so it may impact on the insect status (i.e., heat stress).
376-377 – could different dispersal rates matter more for males vs females? Is the dispersal ability of females more important as they are required to lay the eggs?
- Response: Thank you for the comment. We added a sentence in the end of the paragraph.
Figure A2 – Images could be improved
- Response: Thank you for the comment. Unfortunately, we don’t have high quality images, please understand.
Overall, I don’t think this study needs much, but there were two methods that concerned me that will need clarifying and probably correcting before publication. These are the experiment time sampling from line 147 and part 1 of “Linear mixed models” (see above).
- Response: Thank you for the valuable comments and suggestions. We carefully corrected our manuscript accordingly.